# Topological defects in self-assembled patterns of mesenchymal stromal cells *in vitro* are predictive attributes of condensation and chondrogenesis

**Ekta Makhija[1]\*, Yang Zheng[1,2,3], Jiahao Wang[4], Han Ren Leong[1,5], Rashidah Binte Othman[1], Ee Xien Ng[1], Eng Hin Lee[1,2,3], Lisa Tucker Kellogg[1,6], Yie Hou Lee[1,7,8], Hanry Yu[1,4,9,10], Zhiyong Poon[1,8,11]\*, Krystyn J. Van Vliet[1,12]\***

1 Critical Analytics for Manufacturing Personalized-medicine (CAMP) Interdisciplinary Research Group, Singapore-MIT Alliance for Research and Technology (SMART), Campus for Research Excellence and Technological Enterprise (CREATE), Singapore, Singapore, 2 NUS Tissue Engineering Programme, Life Sciences Institute, National University of Singapore, Singapore, Singapore, 3 Department of Orthopaedic Surgery, National University of Singapore, Singapore, Singapore, 4 Mechanobiology Institute, National University of Singapore, Singapore, Singapore, 5 Engineering Science Programme, College of Design and Engineering, National University of Singapore, Singapore, Singapore, 6 Cancer and Stem Cell Biology, Duke-NUS Medical School, Singapore, Singapore, 7 Obstetrics and Gynaecology Academic Clinical Programme, SingHealth Duke-NUS, Singapore, Singapore, 8 SingHealth Duke-NUS Cell Therapy Centre, Singapore, Singapore, 9 Department of Physiology, National University of Singapore, Singapore, Singapore, 10 Institute of Bioengineering and Bioimaging, Agency for Science, Technology and Research, Singapore, Singapore, 11 Department of Haematology, Singapore General Hospital, Singapore, Singapore, 12 Department of Materials Science and Engineering, Department of Biological Engineering, Massachusetts Institute of Technology, Cambridge, MA, United States of America

\* emakhija@gmail.com (EM); poon.zhi.yong@sgh.com.sg (ZP); krystyn@mit.edu (KJVV)

**Data Availability Statement:** The original raw images from which the results presented in the

## Abstract

Mesenchymal stromal cells (MSCs) are promising therapeutic agents for cartilage regeneration, including the potential of cells to promote chondrogenesis in vivo. However, process development and regulatory approval of MSCs as cell therapy products benefit from facile *in vitro* approaches that can predict potency for a given production run. Current standard *in vitro* approaches include a 21 day 3D differentiation assay followed by quantification of cartilage matrix proteins. We propose a novel biophysical marker that is cell population-based and can be measured from *in vitro* monolayer culture of MSCs. We hypothesized that the self-assembly pattern that emerges from collective-cell behavior would predict chondrogenesis motivated by our observation that certain features in this pattern, namely, topological defects, corresponded to mesenchymal condensations. Indeed, we observed a strong predictive correlation between the degree-of-order of the pattern at day 9 of the monolayer culture and chondrogenic potential later estimated from *in vitro* 3D chondrogenic differentiation at day 21. These findings provide the rationale and the proof-of-concept for using self-assembly patterns to monitor chondrogenic commitment of cell populations. Such correlations across multiple MSC donors and production batches suggest that self-assembly patterns can be used as a candidate biophysical attribute to predict quality and efficacy for MSCs employed therapeutically for cartilage regeneration.

study have been derived are available on Mendeley Data https://doi.org/10.17632/bj28ycg3n2.1.

**Funding:** This research was funded by the National Research Foundation, Prime Minister's Office, Singapore under its Campus for Research Excellence and Technological Enterprise (CREATE) programme, through Singapore MIT Alliance for Research and Technology (SMART): Critical Analytics for Manufacturing Personalised-Medicine (CAMP) Interdisciplinary Research Group. The funders had no role in study design, data collection and analysis, decision to publish, or preparation of the manuscript.

## Introduction

The growing promise of cell therapy for tissue regeneration calls forth a need for rapid and label-free assays that can predict the potential of the cell product to differentiate into desired tissue type. The case considered here is that of mesenchymal stromal cell (MSC) therapy for cartilage repair. Since surgical and pharmaceutical methods of treating cartilage degeneration associated with osteoarthritis restore only limited function, mesenchymal stromal cells that have the potential to differentiate into chondrocytes are of potential translational interest. The approval of such cell-based products by health regulatory authorities anticipates demonstration of identity, safety, purity, and potency of the product. Assays are preferably rapid and label-free, enabling in-line qualification during the cell manufacturing process. However, the current standard in vitro assay for predicting cartilage regeneration potency of MSCs is chondrogenic differentiation, a relatively slow process comprising 21 days of differentiation in 3D pellet using 100K cells, followed by cartilage matrix protein quantification [1]. So far, most biophysical models for predicting MSC differentiation have been based on single-cell features such as cell geometry, actin structure, nucleus geometry, etc. [2–4]. However, MSC differentiation is a cell population-based phenomenon, as suggested by the high confluency requirement of the starting culture in MSC *in vitro* differentiation protocols [5]. A recent study even suggests over-confluence of MSC expansion cultures for effective chondrogenesis, that is conventionally avoided as cell-cell contact inhibits growth [6]. Accordingly, cell population-based features might offer a more direct prediction of the MSC differentiation outcome.

We hypothesized that the self-assembled patterns that emerge in confluent MSC monolayers would be predictive of the chondrogenic potential. This hypothesis was based on the rationale that certain features within the self-assembled pattern corresponded to mesenchymal condensations,–the self-aggregation of mesenchymal cells that is an early and critical step during the development of skeletal tissues [7–9]. The mesenchymal condensations during skeletal development appear as a regularly-spaced pattern of spots that correspond to nodules of the developing limb. Conventionally, computational biologists have modelled the patterning of these condensations via chemotaxis and haptotaxis [10–12].

Interestingly, the self-assembled pattern of cellular swirls that emerges in cell monolayers has been extensively studied in the field of active matter physics, and modelled as kinetic phase transition [13], jamming transition [14], and glass transition [15]. Recently, this swirl pattern has been compared to the turbulence pattern [16, 17], and liquid crystal pattern [18, 19]. The liquid crystal-like pattern along with liquid crystal-like defects in cellular self-assembly has been observed in various biological systems [20, 21] with discoveries that the so-called topological defect sites are the scenes for fundamental morphogenetic events, such as epithelial cell extrusion [22] and neural crest cell migration [23]. In this work, we leverage the relationship between topological defects in cell monolayers and morphogenesis in tissue development to propose an assay and potential attribute predictive of the MSC cell therapy product potency for cartilage repair.

With the rationale that self-assembled patterning precedes morphogenesis, we hypothesized that the quantification of this pattern in confluent MSC monolayers would enable an early and straightforward prediction of chondrogenic potential. Through time-lapsed image analysis of the MSCs in expansion culture conditions (i.e., not chemically induced differentiation conditions) in two-dimensional multi-well formats, we observed a strong correlation between pattern variance at day 9 of cellular swirl assay with the protein level quantified from chondrogenic pellet after 3D chondrogenic differentiation assay at day 21. With development of advanced machine learning algorithms for forecasting the self-assembly patterns as well as in vivo validation, this candidate biophysical attribute can potentially be incorporated as a label-free critical quality attribute during cell manufacturing process.

## Results

### Cellular swirls emerge in confluent cultures of bone marrow-derived MSCs (bm-MSCs)

A visual inspection of any *in vitro* culture on tissue culture plastic of bm-MSCs at confluency shows cellular swirls formed by alignment between neighboring cells (Fig 1A). To observe the origin of these emergent swirls, we performed time-lapse imaging for 2 weeks, starting from sparsely seeded single cells until confluency and emergence of swirls (S1 and S2 Movies). We observed that when local cell density reached a critical value, around day 10 in our movies, the cells transitioned into a collective motion resembling fluid-like behavior. This collective cell flow then produced turbulence-like vortices, building towards the appearance of swirls formed by alignment of neighboring cells.

Motivated by the possibility of using the cellular swirl pattern as a morphological feature for prediction of chondrogenic potential, we generated cellular swirls in bm-MSCs systematically by seeding cells in 48-well plates at a density of 50,000 cells/cm$^2$ for up to 12 days (Fig 1B, see *Discussion* for why this culture vessel and this cell density was chosen). To visualize the cellular swirls, we fixed the cells and stained their actin and nuclei using phalloidin and DAPI respectively because the 48-well plates were not suitable for phase-contrast imaging. Samples were fixed on day 3, 6, 9, or 12 post-seeding for the time study. Images of actin and nuclei were acquired for tiles across the whole well and stitched to capture the complete pattern formed in the well. Before proceeding with quantification of the swirl pattern, we qualitatively examined certain features in the pattern, especially the actin arrangement and cell density at centers of the swirls.

### Cells self-aggregate at swirl centers, the so-called 'positive' topological defects

Zoomed-in actin images of the swirl pattern showed topological defects of 'comet' (+1/2) type and 'spiral' (+1) type. In the DAPI-stained nucleus images, these topological defect sites corresponded to highly dense cell clusters, often with overlapping nuclei (Fig 2 and S1 Fig). These high density cell clusters that appear at swirl centers resemble mesenchymal condensations–the self-aggregation of mesenchymal cells during early stage of cartilage and bone development [7–9]. The reverse was observed at topological defects of the 'tri-radius' (-1/2) type which corresponded to local minima of cell density in the nucleus image (S2 Fig).

### Self-aggregated cell clusters correspond to mesenchymal condensations

To test whether the cells that aggregate at +1/2 and +1 defect sites indeed correspond to mesenchymal condensations, we stained the day 8 samples with fluorescently tagged peanut agglutinin (PNA) antibody, a lectin binding protein used for labelling mesenchymal condensations [24]. Zoom-in images show high intensity of PNA at the cell clusters (Fig 3). Further, immunostaining of transcription factors YAP, SOX9, and RUNX2 showed high intensity in the nuclei within the cell aggregates (S3 Fig).

### Swirl pattern is cell batch-specific and reproducible

We performed a qualitative comparison of the swirl patterns across the phase contrast images of confluent MSCs collected from cultures of various bm-MSC donors maintained by different researchers in our lab. We observed qualitative similarities in the patterns made by the same donor across images captured by different researchers in different months (of 80–90% confluent culture on day 10–12 post seeding with an initial density of 1500–2000 cells/cm$^2$) (S4 Fig).

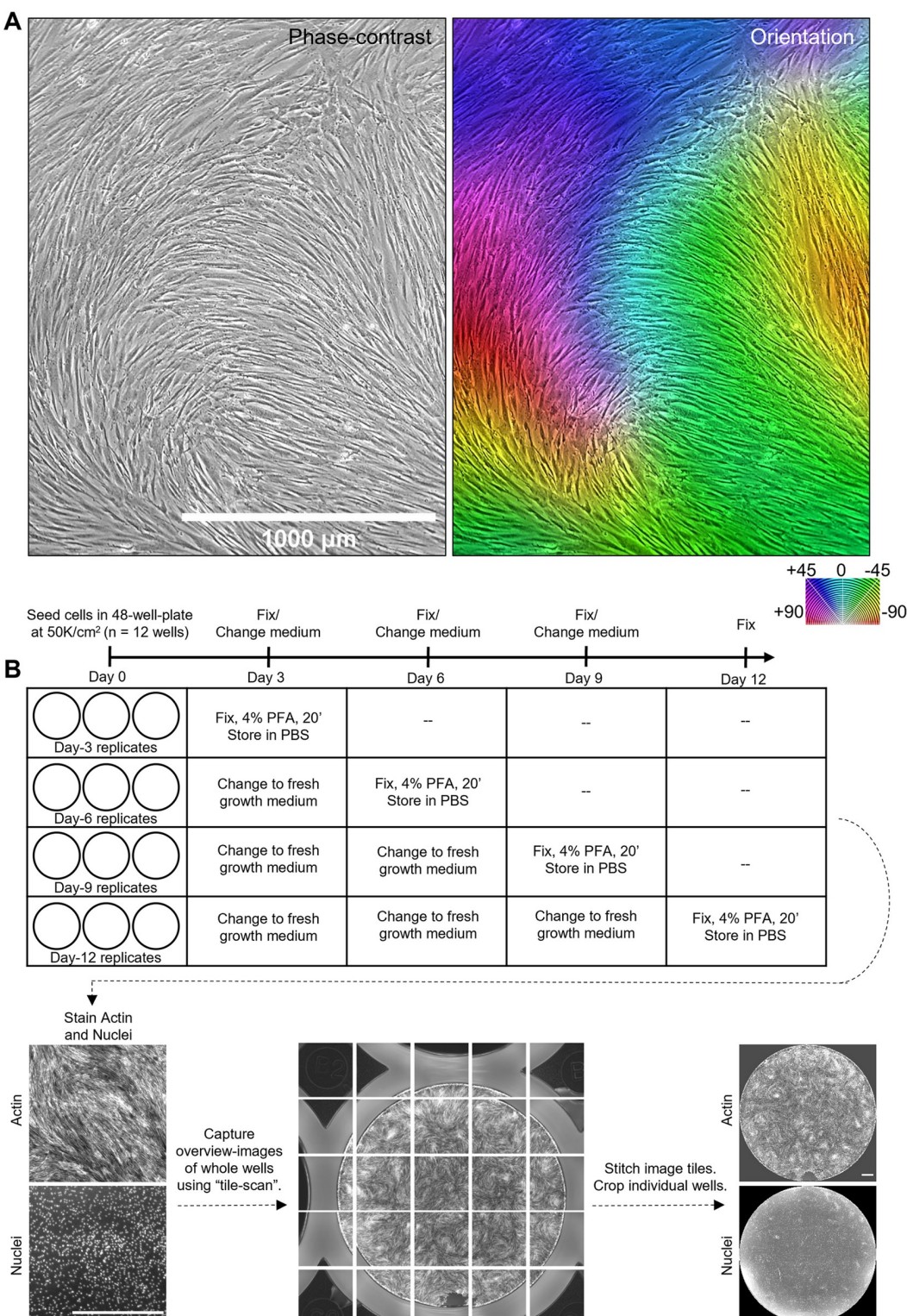

**Fig 1. Cellular swirls emerge in confluent cultures of mesenchymal stromal cells.** (A) Left panel shows phase-contrast image of bone marrow-derived mesenchymal stromal cells cultured to confluency in T75 flasks. Right panel shows its corresponding orientation image generated using OrientationJ plugin in ImageJ. (B) Experiment methodology showing cell sample preparation, staining, imaging, and image processing to generate whole well pattern images. All scale bars are 1000 μm.

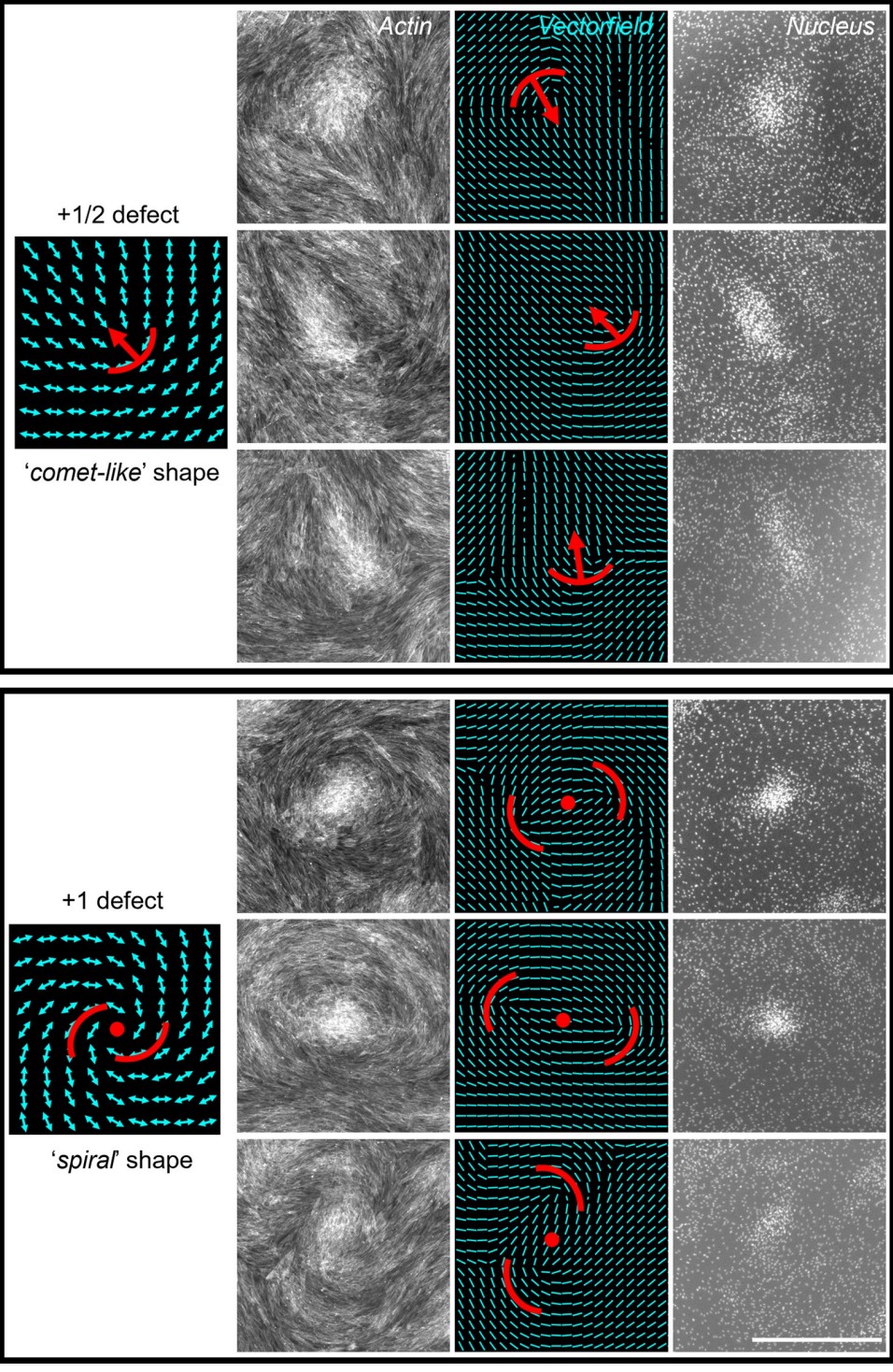

**Fig 2. Cells aggregate at +1/2 and +1 topological defect sites.** Six regions (cropped from whole-well stitched images) showing that cells aggregate at sites of +1/2 and +1 topological defects. The aggregates are visible in the nucleus images, while the defects are visible in the actin and corresponding orientation vectorfield images. Orientation vectorfields were generated using the OrientationJ plugin in ImageJ (see Methods). Scale bar 1000 μm.

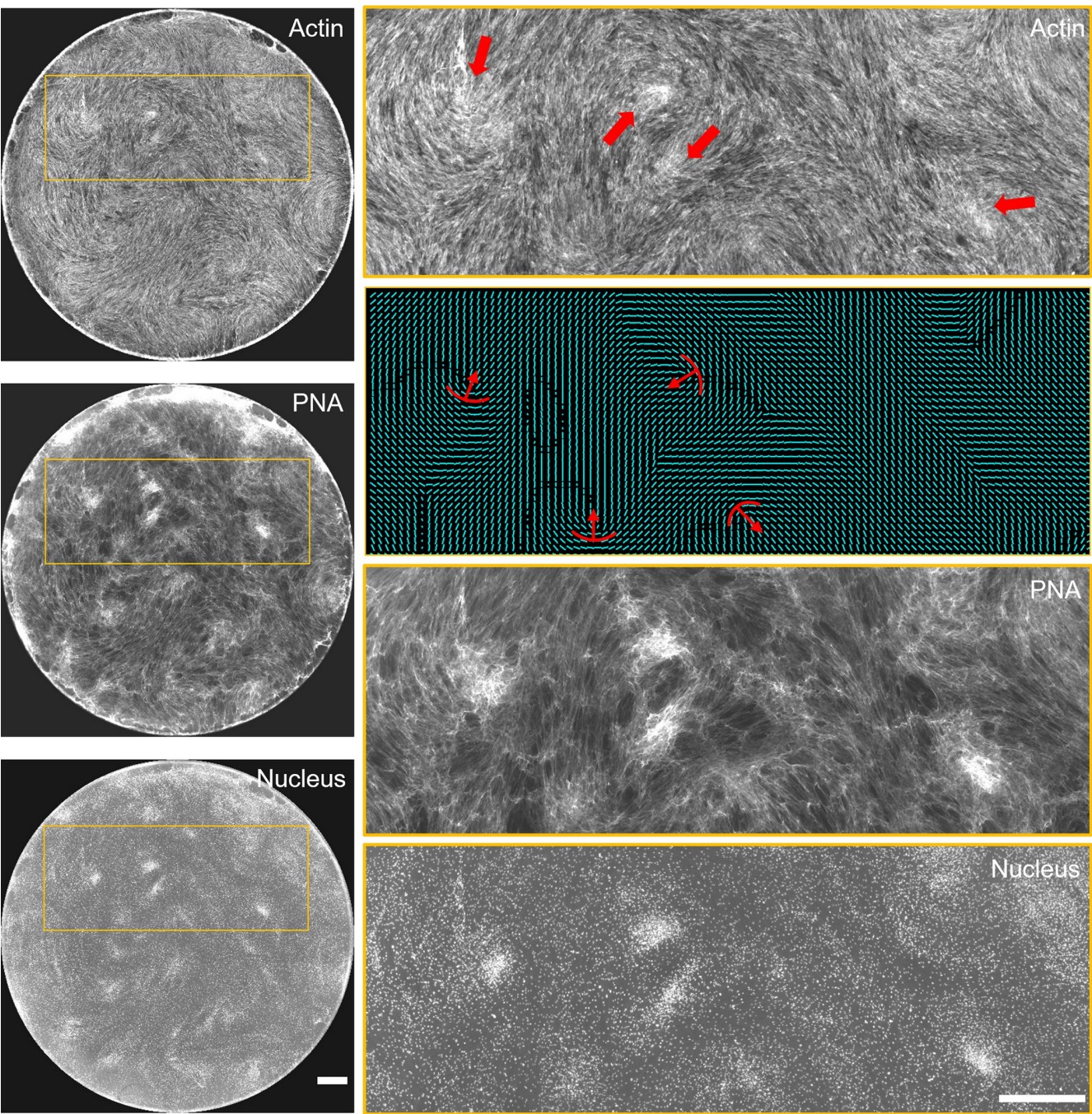

**Fig 3. Cell aggregates at topological defects correspond to mesenchymal condensations.** The cell aggregates arising at defect sites of actin nematic pattern colocalize with peanut agglutinin (PNA), the marker for mesenchymal condensations—a pre-requisite for formation of cartilage or bone during skeletal development. Scale bar 1000 μm.

This observation motivated us to hypothesize that the swirl patterns could be used as a biometric identification marker of the cell batch. To test this hypothesis, we generated and imaged cellular swirls systematically (see Methods) in 3 or more technical replicates across multiple donors. Again, a visual qualitative comparison of the actin images showed that swirl patterns are cell-batch specific and reproducible across technical replicates (Fig 4). Next, we tested

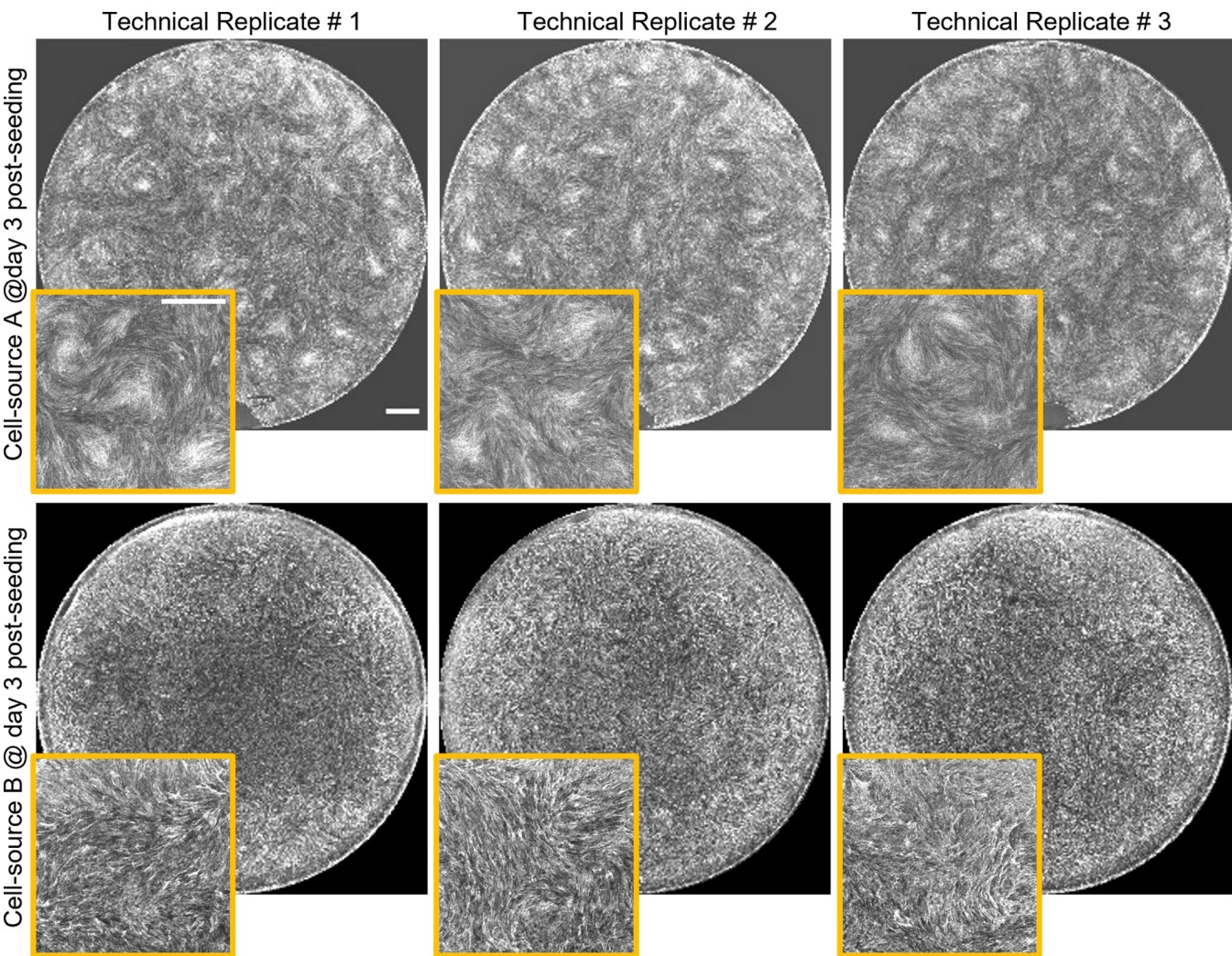

**Fig 4. Self-assembled swirl pattern is cell batch-specific.** Whole-well actin images of cellular swirls generated from two cell sources on day 3 post-seeding. The small-panels show zoomed-in regions cropped from the whole-well actin pattern. Cell source A = donor 1 passage 3, cell source B = donor 2 passage 3. Cells were seeded at 50,000 cells / cm2 for cell sources. Samples for both cell sources were fixed and stained on day 3 post seeding. Scale bar 1000 μm.

whether a quantification of this swirl pattern across various donors correlates with their *in vitro* chondrogenic differentiation potential, since cellular swirls lead to mesenchymal condensations, a crucial step that precedes chondrogenic differentiation.

### Swirl pattern quantification correlates with *in vitro* chondrogenic differentiation potential

To find an image-based early marker of chondrogenic differentiation potential in MSCs, first, the chondrogenic differentiation assay was conducted for 5 donors (see Methods). Briefly, this assay involved 3 week 3D pellet culture in chondrogenic differentiation condition, followed by quantification of cartilage matrix proteins sulfated glycosaminoglycans (sGAGs) and collagen-II (Col2). Simultaneously, for generating cellular swirls, cells from the same 5 donors were cultured at 50,000 cells/cm$^2$ in 48-well plates and stained for nucleus and actin at days 3,6,9, and 12 post seeding. For image quantification, two different methods were adopted; one using nucleus image to directly quantify the mesenchymal condensations, and the other using actin

image to indirectly quantify the condensations as topological defects in the swirl pattern. In the first method, the total area of cell clusters was quantified, for which the nucleus images were thresholded, and the total area of segmented regions was measured (S5A and S5B Fig, see Methods). While the total area of clusters per well increased from day 3 to day 12 for all donors (S5C Fig), it did not correlate strongly with the chondrogenic differentiation quantified using cartilage matrix protein expression (Pearson's correlation coefficient r < 0.5). In the second method, swirl pattern was quantified from actin orientation, as we had observed that topological defects in swirl pattern correspond to mesenchymal condensations (Fig 3)–an early marker of chondrogenic differentiation. Hence, we explored a quantification method that would differentiate the actin organization at topological defect from the other regions having fully ordered or disordered actin. For the purpose of illustration, we cropped from whole-well actin images, three small regions corresponding to each type of pattern: 'topological defect', 'disorder', and 'order' (Fig 5A). The correspondingly cropped nucleus images (S6 Fig) confirmed occurrence of condensation only for the topological defect region. Next, the orientation and coherency images were generated from the actin images for the three regions using the open source plugin OrientationJ for Fiji (see Methods). The orientation shows the direction of local features, and varies from -90 to +90 degrees. The coherency measures the local structure: it is 1 when the local structure has a dominant orientation and 0 when the local structure is unaligned or isotropic. Subsequent computation of the variance of coherency (VoC) for the three regions showed that the topological defect region has higher VoC compared to regions that are disordered or highly ordered.

Following this method, VoC was computed from all whole-well actin images (5 donors x 4 time points x 3 technical replicates = 60 whole-well actin images) (see Methods). The quantity VoC showed high technical reproducibility as suggested by a low coefficient of variation (less than 30%) for all donors (S7 Fig). Next, we tested the correlation of VoC with cartilage matrix protein quantification across 5 donors at the 4 time points: day 3, 6, 9, and 12. Interestingly, the VoC at day 9 showed strong correlation with both cartilage matrix components, generating a Pearson's correlation coefficient r = 0.88 for sGAG, and r = 0.98 for Col2 (Fig 5B), while the VoC at earlier time points did not correlate strongly with the differentiation markers (S8 Fig). Next, we tested whether the quantity VoC could be measured from phase-contrast images of living (unfixed) cells, as this would have implications for in-line label-free monitoring of chondrogenic differentiation efficiency. We observed that the coherency histogram obtained from actin image matches that obtained from phase-contrast image for the same region (S9 Fig).

## Discussion

The use of mesenchymal stromal cell therapy for cartilage repair requires an assessment of the efficacy of the MSC product to repair the cartilage tissue. The current *in vitro* assay that assesses the cartilage formation efficacy of MSCs is the chondrogenic differentiation assay comprising of 21 days of 3D pellet culture followed by quantification of cartilage matrix proteins. There is a need for efficacy prediction assays that enable early prediction and are non-destructive, so they can be incorporated as in-line assays within the cell manufacturing pipeline. Our work provides proof-of-concept that the pattern of self-assembly in mesenchymal stromal cells can predict their chondrogenic potential in 9 days with scope for an even earlier and label-free prediction.

We hypothesized that cellular swirl patterns emerging on tissue culture plastic from collective self-organization of cells would be predictive of the chondrogenic differentiation potential of the batch of cells since such self-organization is known to precede tissue morphogenesis. To test this, we first generated the self-assembled cellular swirls in confluent human bone

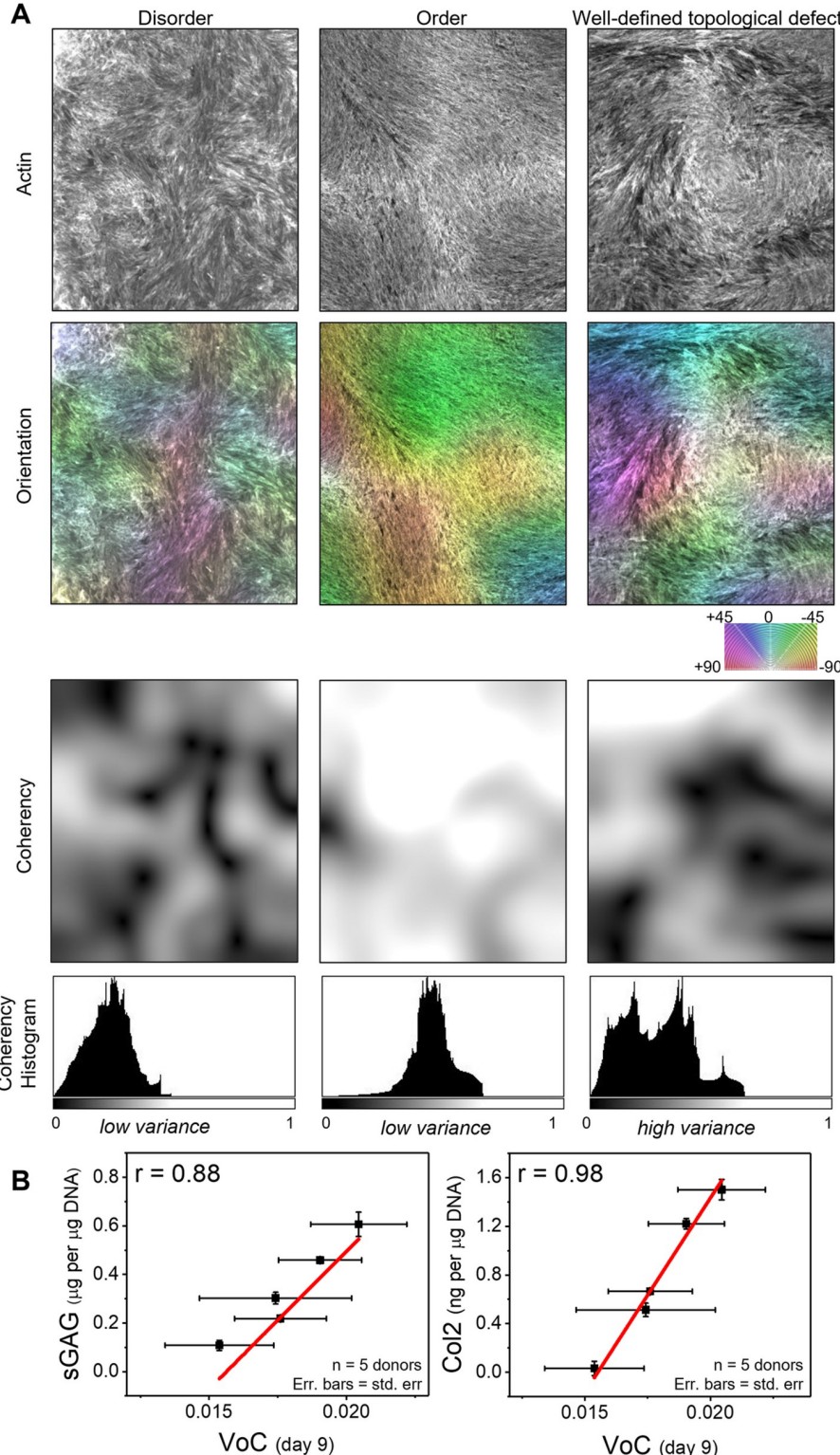

**Fig 5. Variance of pattern coherency correlates with *in vitro* chondrogenic differentiation.** (A) The first row shows actin images corresponding to three regions where the pattern may be classified as having '*disorder*', '*order*', and '*topological defect*'. The second and the third rows show orientation and coherency images for the three regions, generated from the corresponding actin images using OrientationJ (see Methods). The values of orientation vary from -90 to +90 degrees, and the values of coherency vary from 0 to 1. The last row shows histogram of coherency images.

Scalebar 1000 μm. (B) XY scatter plots, where X = variance of coherency (VoC) calculated from day 9 patterns; and Y = expression level of cartilage matrix proteins sGAG and Col2 quantified from the 21 day chondrogenesis differentiation assay. The dots correspond to mean of 3 technical replicates, while the X and Y error bars correspond to standard error. Red line shows the linear fitting.

marrow-derived MSCs by controlling seeding density, culture vessel geometry, and culture duration. The value of starting cell density was chosen such that cells are close to 90% confluent upon attachment, as this would minimize the time required for emergence of cellular swirls. The culture vessel chosen was 48-well plate, so as to optimize between the number of swirl features per well and the byte-size of the whole-well overview image. Recent studies of collective cell behavior in field of active matter physics suggest that the patterning observed in biological tissues resembles the patterns and topological defects observed in nematic liquid crystals [18, 19]. A salient feature of the nematic liquid crystal pattern is topological defects, i.e. the sites within the pattern where sudden changes in orientation occur, for example, the centers of the spiral swirls [25]. We were able to identify positive and negative, integer and half-integer types of topological defects within the patterns visible in actin-stained images of the cellular swirls. We observed self-aggregated cell clusters located at the so-called 'comet' (+1/2) and 'spiral' (+1) topological defect sites within the nematic pattern. Such clustering of particles at positively charged topological defect sites can be induced by turbulence-like flow in active systems as is recently being explored [26]. These cell clusters were identified as mesenchymal condensations, the early stage of cartilage formation, via labelling with peanut agglutinin antibody. Further, we observed high expression of transcription factors YAP, SOX9, and RUNX2 in the cells within the clusters. This suggests that the high mechanical stresses, known to arise at +1/2 and +1 defect sites, possibly activate the mechanosensitive transcription factor YAP in the mesenchymal condensations, followed by downstream activation of RUNX2 and SOX9, the master transcription factors of osteogenic and chondrogenic differentiation [27].

Interestingly, we observed that the swirl patterns made by cells obtained from different donors were cell batch-specific and reproducible across technical replicates. Next, we explored the possibility of using swirl pattern to predict the chondrogenic differentiation potential of MSCs. Toward this, we adopted two methods of pattern quantification–using nucleus image to quantify the total area of mesenchymal condensations or cell clusters, and using the actin image to quantify the topological defects. The first quantity, i.e. total area of cell clusters computed from nucleus image did not correlate strongly with chondrogenic differentiation markers. This could have two possible implications: (1) higher total area of clusters does not necessarily mean better chondrogenesis, or (2) the total area computed from intensity-based thresholding does not successfully capture the quality of mesenchymal condensations. The second quantity, computed from actin images, captured the level of order in the actin orientation. This quantity, 'coherency', ranges from 0 to 1. It is uniformly high in regions that have fully ordered actin fiber arrangement, uniformly low in regions that have for fully disordered arrangement, and is spread from low to high in regions that have a topological defect. Thus, we computed the 'variance of coherency' (VoC) as a measure of the topological defects in the actin image. The VoC is high for regions corresponding to defect and low for fully ordered or fully disordered regions. Interestingly, the VoC of day 9 swirl pattern correlated strongly with the cartilage matrix proteins expressed on day 21 of the chondrogenic differentiation assay (n = 5 donors). The low correlation of VoC at earlier time points (VoC of day 3 and day 6) with the differentiation markers is possibly because of lag in emergence of cellular swirls across donors, since differences in cell morphology and proliferation rates would affect the value of critical density at which the swirls emerge and the time required to achieve that critical density.

To summarize, we showed that swirl-pattern quantification could be used as a potential critical quality attribute for predicting efficacy of mesenchymal stromal cell product for cartilage repair. Here, we do not claim the patterns to be donor-specific, or inherently genetic, but rather cell preparation condition-specific. We can refer to this as batch specific. This distinction is important because MSC phenotype is established to be easily moderated by culture conditions, and a change in those conditions could alter both the observed VoC and chondrogenic potential of that cell batch. Further aspects regarding implications of swirl-pattern quantification for cell therapy manufacturing are discussed below.

A crucial stage during the cell therapy manufacturing is that of cell expansion. Typical protocol optimized for expansion recommends seeding mesenchymal stromal cells sparsely and harvesting them before they reach confluency [28, 29]. Hence, a prediction model based on single-cell morphological features could be easily incorporated in-line within the manufacturing pipeline. While cellular swirls only appear at high confluency, advances in forecasting of active nematics [30] from morphology, migration rate, and proliferation rate of single-cells could pave the way for an earlier prediction from lower confluency of ~80%, typical of the MSC harvest stage of the cell therapy manufacturing. Secondly, with the new understanding that cellular swirls lead to mesenchymal condensations, cell manufacturers for cartilage regeneration could consider culturing MSCs to near 100% confluency to prime the cells towards chondrogenesis. In such a case, an in-line label-free monitoring of chondrogenic differentiation efficacy via swirl pattern would be feasible, as we have shown that VoC can also be measured from phase-contrast images of living (unfixed) cells. Thirdly, culturing MSCs on engineered patterned surfaces that induce formation of topological defects [31] would provide a better control for manufacturing desired cell therapy product.

Growing evidence from *in vivo* cartilage repair studies shows that the implanted MSCs play an indirect role in tissue repair via secretion of paracrine factors as opposed to directly differentiating into chondrocytes [32]. While our current work shows that collective cell patterns provide an early prediction of the *in vitro* chondrogenic differentiation potential, future studies can establish whether these patterns also predict the *in vivo* cartilage repair outcome.

## Materials and methods

### Cell-source and cell expansion

Human adult bone marrow-derived MSCs (bm-MSCs) from 5 different donors (males, age 20–30 years) were purchased from Lonza and StemCell Technology. The first round of expansion for all donors from passage 1 to passage 2 was done by a single researcher. Cells were seeded in culture flasks (ThermoFisher Nunc) at 2000 cells/cm$^2$ using growth medium comprising of Dulbecco's modified Eagle's medium, low glucose and pyruvate (Gibco), supplemented with 10% MSC certified fetal bovine serum (Gibco) and 1% penicillin and streptomycin (Gibco). Medium was changed every 3 days until the cells reached ~80% confluency, typically 1–2 weeks, when the cells were harvested using the trypsin (Gibco). The second passage cells were then stored in common repository by cryo-preserving in batches of 3.5*10$^5$ per vial in freezing medium comprising of growth medium with 10% DMSO (Sigma). For subsequent use in various experiments, different researchers typically expanded the MSCs from passage 2 to passage 3 or passage 4 following the same expansion protocol as above. For generating cellular swirls in 48-well plates in the current study, second passage vials of all 5 donors were first thawed, and serial expansion was performed until a cell-count of 7*10$^5$ cells was achieved for each donor (~0.6*10$^5$ cells/well x 3 technical replicates x 4 time-points). Typically, this cell count was achieved at the third passage for fast-growing donors and fourth passage for slow-growing donors.

## Sample preparation for generating cellular swirls

Cells were seeded in 48-well plates at a density of 50,000 cells /cm$^2$. Note that the first and last rows as well as the first and last columns of the well-plate were not used for seeding because limitations in XY translation of the microscope stage affects the whole-well imaging for these wells. Also, samples corresponding to different time-points were seeded in different well-plates for the reason that paraformaldehyde (PFA) vapors from the samples being fixed resulted in cell death in other samples within the same well plate. For the time-study experiments, the samples were fixed on either day 3, 6, 9, or 12 with medium change every 3 days until they're fixed. For fixing, growth medium was replaced with 4% PFA in PBS (Biotium, 200ul per well) for 20 minutes, following which the PFA was replaced with PBS. The fixed samples were stored in the incubator or refrigerator until all time points were fixed and ready for staining.

## Fluorescent staining

Fixed samples were first permeabilized using 0.1% triton X-100 detergent solution (Thermo Scientific) for 5 minutes, followed either by blocking and primary antibody staining or directly by 30 minute staining of nucleus (NucBlue Fixed Cell ReadyProbes, Life Technologies) and actin (ActinGreen 488 ReadyProbes, Life Technologies).

For antibody staining, permeabilization step was followed by an hour of incubation with blocking buffer comprising 1% bovine serum albumin solution (Sigma) with 0.1% tween (Sigma). Next, the primary antibody, diluted in blocking buffer, was added and samples were incubated overnight at 37˚C. Peanut agglutinin (lectin PNA from Arachis hypogaea, Alexa Fluor 488 Conjugate, Life Technologies, dilution 1:20), SOX9 (rabbit monoclonal to SOX9 Alexa Fluor 647, Abcam, dilution 1:50), RUNX2 (rabbit monoclonal RUNX2, Alexa Fluor 647, Abcam, dilution 1:50), and YAP (rabbit monoclonal to active YAP1 Alexa Fluor 488, Abcam, dilution 1:50) antibody staining were performed in day 8, day 9, or day 10 samples.

## *In vitro* 3D chondrogenic differentiation assay

MSCs were centrifuged down into 3D pellet of $1 \times 10^5$ cells in 96 well non-adhesive plates (Nest Biotechnology Co., Ltd.). Cells were cultured in chondrogenic differentiation medium, which consists of high glucose DMEM supplemented with $10^{-7}$M dexamethasone, 1% ITS + premix, 50 μg/ml ascorbic acid, 1 mM sodium pyruvate, 0.4 mM proline and 10 ng/ml TGF-β3 (R&D Systems, Minneapolis, MN). Differentiation was carried out for 3 weeks, with complete medium exchange every 3 days.

## Sulfated glycosaminoglycan (sGAG) and type II collagen quantification

Samples were digested with 10 mg/mL of pepsin in 0.05 M acetic acid at 4˚C, followed by digestion with elastase (1 mg/mL). The amount of sulfated glycosaminoglycan (sGAG) was quantified using Blyscan sGAG assay kit (Biocolor, UK) according to manufacturer's protocol. Absorbance was measured at 656 nm and sGAG concentration (μg per μg DNA) was extrapolated from a standard curve generated using a sGAG standard. Type II Collagen (Col 2) content was measured using a captured enzyme-linked immunosorbent assay (Chondrex, Redmond, WA). Absorbance at 490 nm was measured and the concentration of Col 2 (ng per μg DNA) was extrapolated from a standard curve generated using a Col 2 standard. The amount of both sGAG and Col 2 content were normalized to the DNA content of respective samples, measured using Picogreen dsDNA assay (Molecular Probes, OR, USA). Three replicates were analyzed within each group.

## Static imaging and image pre-processing

Whole-well overview images of actin, nucleus, and other fluorescently-labelled proteins (PNA, SOX9, RUNX2, YAP) were captured on Olympus IX83 fluorescence microscope using 4X objective and tile-scan option in Metamorph software. Individual tiles were 2048 x 2048 pixels, pixel size was 1.6 microns. Image tiles were stitched using the "Stitching" plugin in ImageJ [33]. Individual wells were then cropped manually from the stitched images using the circular cropping tool in ImageJ (radius 3636 pixels). Note that the stitching often generated artefact in the form of high intensity at the junctions between neighboring tiles. However, we observed that these intensity artefacts did not affect quantification of coherency from whole-well actin images.

## Quantification of collective-cell pattern from nucleus images

Nucleus images of individual wells were imported into the MATLAB Image Processing Toolbox (Mathworks). First, these images were cropped with a circle mask (radius 2136 pixels) to eliminate the influence caused by uneven light in the corner of wells and then adjusted by contrast enhancement using contrast limited adaptive histogram equalization [34]. For noise removal, two binary masks were generated separately via image global threshold and local adaptive threshold, and then used together to subtract the background. De-noised nucleus images were first processed with a Gaussian filter (radius 100 pixels) to obtain the nucleus distribution heatmap. This heatmap was then normalized and transformed into a binary mask using a threshold of 0.5. Total cell aggregate area per well was estimated from this binary mask.

## Identifying various topological defects from actin images

Topological defects in swirl pattern are regions where there is irregularity in the orientation of actin. The defect sites where the orientations of neighboring actin vectors make a comet-like shape are called +1/2 defects (Fig 2), where they make a spiral shape are called +1 defects (Fig 2), and where they are splayed out are called -1/2 defects (S2 Fig).

## Quantification of collective-cell pattern from actin images

Orientation and coherency images were generated from whole-well actin images in ImageJ via the plugin OrientationJ [35], with a local window size of 100 pixels and 'Gaussian' gradient-type. Coherency images were cropped manually using the circular cropping tool in ImageJ (radius 3336 pixels) to remove the edge effects. Histogram of the coherency image was generated in ImageJ using Analyze → Histogram. The values of variance were read-out from the histograms.

## Live cell time-lapse imaging

14-day time-lapse imaging to visualize the transition from single cell to cellular swirls was performed as described in an earlier work [36]. Briefly, MSCs were seeded very sparsely at 100 cells / cm$^2$ in a glass bottom petri dish, which was mounted on the microscope stage-top incubator. Phase-contrast images were captured using the 10X objective every fifteen minutes. Initially, tile images in a 2x2 grid were captured around the single cells. As the cells migrated and proliferated, the number of tiles was increased to 3x3 on day 4, 4x4 on day 8, and 5x5 on day 12 of the time-lapse imaging.

## Supporting information

**S1 Fig. Additional images of cell aggregation at +1/2 and +1 topological defects.** Five regions (cropped from whole-well stitched images) showing that cells aggregate at sites of +1/2 and +1 topological defects. The aggregates are visible in the nucleus images, while the defects are visible in the actin and corresponding orientation vectorfield images. Orientation vector-fields were generated using the OrientationJ plugin in ImageJ (see Methods). Scale bar 1000 μm.
(TIF)

**S2 Fig. -1/2 topological defect sites have local minimum in cell density.** Three regions (cropped from whole-well stitched images) showing that cells recede from -1/2 topological defects. The defects are visible in the actin and corresponding orientation vectorfield images, while the density-minima are visible in the nucleus density colormap. Orientation vectorfields were generated using the OrientationJ plugin in ImageJ (see Methods). Nucleus density color-maps were generated using an averaging filter (100 pixels, i.e., 16 μm) on the nucleus intensity images in MATLAB. Scale bar 1000 μm.
(TIF)

**S3 Fig. Upregulation of SOX9, RUNX2, and YAP in cell aggregates at defect sites.** Regions cropped from whole-well stitched actin, nucleus, and transcription factor images show higher levels of SOX9, RUNX2, and YAP in the nuclei of cells aggregated at the topological defect sites. SOX9, RUNX2, and YAP were stained in day 10, day 9, and day 8 samples respectively. Scale bar 1000 μm.
(TIF)

**S4 Fig. Similarity in self-assembled patterns captured by different researchers.** Phase-contrast images of bm-MSCs at confluency captured by two different researchers on day 10–12 post seeding with 1500–2000 cells/cm$^2$ starting density. Scale bar 1000 μm.
(TIF)

**S5 Fig. Temporal dynamics of total cell aggregate area.** (A) Whole-well stitched nuclei images from a single donor at time 3, 6, 9, 12 days post seeding. (B) Binary masks generated by thresholding the above nuclei images. Spots corresponding to debris were manually removed. (C) Total cell aggregate area per well plotted as a function of time for 5 MSC donors (n = 3 technical replicates per donor). (D) Pearson correlation coefficient for day 3, 6, 9, 12 total area vs levels of matrix protein, n = 5 donors. Scale bar 1000 μm.
(TIF)

**S6 Fig. Condensations do not appear at disordered or highly ordered regions.** The first row shows actin images corresponding to three regions where the pattern may be classified as having '*disorder*', '*order*', and '*topological defect*'. The second and the third rows show the corresponding nucleus and vectorfield images for the regions. Scale bar 1000 μm.
(TIF)

**S7 Fig. Technical reproducibility of variance of coherency (VoC).** Coefficient of variation (%), computed as standard deviation / mean across 3 technical replicates for each of the 5 donors.
(TIF)

**S8 Fig. Variance of pattern coherency correlates with *in vitro* chondrogenic differentiation.** Pearson correlation coefficient for day 3, 6, 9, 12 VoC vs levels of matrix protein, n = 5 donors.
(TIF)

**S9 Fig. Label-free quantification of orientation and coherency.** Comparison of orientation analysis for three different regions shows similarity in orientation, coherency and its variance from actin vs phase-contrast images. Scale bar 1000 μm.
(TIF)

**S1 Movie. Single cell to cellular swirls.** This is a 14-days long time-lapse phase-contrast imaging showing emergence of cellular swirls. Yellow box outlined on the left corresponds to the zoomed region shown on the right.
(AVI)

**S2 Movie. Critical density to cellular swirls.** Time lapse of the last 3 days sub-stacked from S1 Movie.
(AVI)

## Acknowledgments

We thank George Barbastathis, Irmgard Bischofberger, Jongyoon Han, Tetsuya Hiraiwa, and Yusuke Toyama for useful discussions. We are grateful to Jean-Francois Rupprecht for reading the manuscript and providing valuable input.

## Author Contributions

**Conceptualization:** Ekta Makhija, Yang Zheng, Eng Hin Lee, Lisa Tucker Kellogg, Yie Hou Lee, Hanry Yu, Zhiyong Poon, Krystyn J. Van Vliet.

**Formal analysis:** Ekta Makhija.

**Investigation:** Ekta Makhija, Yang Zheng, Jiahao Wang, Han Ren Leong, Rashidah Binte Othman, Ee Xien Ng.

**Methodology:** Ekta Makhija, Yang Zheng, Jiahao Wang.

**Software:** Ekta Makhija, Jiahao Wang.

**Supervision:** Zhiyong Poon, Krystyn J. Van Vliet.

**Visualization:** Ekta Makhija.

**Writing – original draft:** Ekta Makhija, Yang Zheng, Jiahao Wang, Rashidah Binte Othman.

**Writing – review & editing:** Ekta Makhija, Yang Zheng, Lisa Tucker Kellogg, Yie Hou Lee, Zhiyong Poon, Krystyn J. Van Vliet.

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
