## [Decision Letter · Decision Letter 0]

12 Sep 2023

PONE-D-23-21148Topological defects in self-assembled patterns of mesenchymal stromal cells in vitro are predictive attributes of condensation and chondrogenesisPLOS ONE

Dear Dr. Van Vliet,

Thank you for submitting your manuscript to PLOS ONE. After careful consideration, we feel that it has merit but does not fully meet PLOS ONE’s publication criteria as it currently stands. Therefore, we invite you to submit a revised version of the manuscript that addresses the points raised during the review process.

We look forward to receiving your revised manuscript.

Kind regards,

Nileshkumar Dubey

Academic Editor

PLOS ONE

“This research was funded by the National Research Foundation, Prime Minister’s Office, Singapore under its Campus for Research Excellence and Technological Enterprise (CREATE) programme, through Singapore MIT Alliance for Research and Technology (SMART): Critical Analytics for Manufacturing Personalised-Medicine (CAMP) Interdisciplinary Research Group.”

“This research was funded by the National Research Foundation, Prime Minister’s Office, Singapore under its Campus for Research Excellence and Technological Enterprise (CREATE) programme, through Singapore MIT Alliance for Research and Technology (SMART): Critical Analytics for Manufacturing Personalised-Medicine (CAMP) Interdisciplinary Research Group. We thank George Barbastathis, Irmgard Bischofberger, Jongyoon Han, Tetsuya Hiraiwa, and Yusuke Toyama for useful discussions. We are grateful to Jean-Francois Rupprecht for reading the manuscript and providing valuable input.”

“This research was funded by the National Research Foundation, Prime Minister’s Office, Singapore under its Campus for Research Excellence and Technological Enterprise (CREATE) programme, through Singapore MIT Alliance for Research and Technology (SMART): Critical Analytics for Manufacturing Personalised-Medicine (CAMP) Interdisciplinary Research Group.”

4. We noted in your submission details that a portion of your manuscript may have been presented or published elsewhere. [Preprint on BioRxiv

https://doi.org/10.1101/2022.05.30.493944] Please clarify whether this [conference proceeding or publication] was peer-reviewed and formally published. If this work was previously peer-reviewed and published, in the cover letter please provide the reason that this work does not constitute dual publication and should be included in the current manuscript.

Reviewers' comments:

Reviewer's Responses to Questions

**Comments to the Author**

1. Is the manuscript technically sound, and do the data support the conclusions?

Reviewer #1: Partly

Reviewer #2: Partly

2. Has the statistical analysis been performed appropriately and rigorously? 

Reviewer #1: No

Reviewer #2: No

3. Have the authors made all data underlying the findings in their manuscript fully available?

Reviewer #1: Yes

Reviewer #2: Yes

4. Is the manuscript presented in an intelligible fashion and written in standard English?

Reviewer #1: Yes

Reviewer #2: No

5. Review Comments to the Author

Reviewer #1: The authors presented an in vitro method to predict the chondrogenic potency of MSCs. The authors examed the self-assembled patterns of confluent BM-MSCs and correlated these patterns with cartilage ECM productions. In general, the authors described a novel and easy method that predicts the chondrogenesis of MSC. However, there are some issues the authors should address.

The results section consists of a mixture of presenting and discussing the results. Please only present the results in the result section.

Line 166 - 168, quantitative data are required to support the claim.

Line 171 - 209, Figure S5 quantifies the number of clusters using nuclei count. Figure 5 quantifies the VoC of different patterns. What is the difference between the number of clusters and VoC? Is there a specific reason why the author changes the quantification method for Figure S5 and figure 5?

Line 192 - 193. Please present merged nuclei and Actin fluorescence images of the whole well to support the claim of "occurrence of condensation only for the topological defect region".

Line 256 - 257, the results of Fig S8 should be presented in the result section, not in the discussion section.

Line 267 - 268, the sex and age of the donors should be described.

In the method section, the authors should describe how they define different patterns: ordered, disordered, topological defects, and +1/2, +1 defects.

Reviewer #2: In this manuscript, the Authors have set up an algorithm for computational analysis of cell expansion in monolayer cultures. Despite the great novelty, several issues should be addressed.

1. Monolayer cultures of MSCs can be used for chondrogenesis; however, other methods have been widely described and adopted, such as spheroids and 3D high-density cultures, with a more efficient chondrogenesis (e.g., Lamparelli EP, Ciardulli MC, Giudice V, Scala P, Vitolo R, Dale TP, Selleri C, Forsyth NR, Maffulli N, Della Porta G. 3D in-vitro cultures of human bone marrow and Wharton's jelly derived mesenchymal stromal cells show high chondrogenic potential. Front Bioeng Biotechnol. 2022 Sep 26;10:986310. doi: 10.3389/fbioe.2022.986310).

2. It is not well discussed how confluency in monolayer culture can be linked to effective chondrogenesis, or how actin expression and orientation can be a good marker of effective chondrogenesis, while results from chondrogenic-related proteins has been added only in supplementary files without any discussion. Moreover, chondrogenesis is triggered by appropriate stimuli in culture medium and requires several days (at least 15 days), while staining and cultures end at day 10 or 12.

3. No data on effective chondrogenesis are reported, and no data on exclusion of different lineage differentiation or fibrotic shift are shown.

4. In vitro chondrogenic differentiation assay conditions are different from those used for setting up the method.

5. A negative control without chondrogenic medium or with a different differentiation medium is not described.

6. Order of references should be corrected and abbreviations should be always used once defined.

7. Major English check should be performed.

6. PLOS authors have the option to publish the peer review history of their article (what does this mean?). If published, this will include your full peer review and any attached files.

Reviewer #1: No

Reviewer #2: No

---

## [Author Response · Author response to Decision Letter 0]

10 Jan 2024

We thank the reviewers and the editor for reviewing our manuscript and providing valuable comments for improving it. Based on these comments, we have made several changes in the manuscript and added a new supplementary figure (Fig. S7). A list of these changes is attached below. A point-by-point response to the reviewers’ comments as well as revised manuscript with and without the tracked changes has been uploaded on the portal. We hope that our revised manuscript meets the journal’s publication criteria.

List of changes in the revised manuscript:

1. Moved several sentences from results section to discussion section (suggested by reviewer#1).

2. Moved the result of supplementary figure S8 from discussion section to results section (suggested by reviewer#1).

3. Added a new supplementary figure S7 to quantify the reproducibility of pattern quantification across technical replicates (suggested by reviewer#1). Also updated figure legends accordingly. 

4. Revised the text in the results sub-section corresponding to pattern quantification to clarify the two methods adopted for quantification that are depicted in Fig. S5 and Fig. 5 (suggested by reviewer#1).

5. Added information about the sex and age of donors in the methods section (suggested by reviewer#1).

6. Added a new sub-section under the methods section to define the various types of topological defects (suggested by reviewer#1).

7. Added the term “3D” at every mention of the in vitro chondrogenic differentiation assay to avoid confusion between this assay vs the monolayer culture for predicting chondrogenesis from self-assembly pattern (based on reviewer#2’s comments).

8. Added 4 lines in the introduction section for literature review on relation between monolayer confluency and effective chondrogenesis (based on reviewer#2’s comments). 

9. Made English corrections and split large sentences into smaller ones (suggested by reviewer#2).

10. Corrected the order of references (suggested by reviewer#2).

---

## [Editor Report · Decision Letter 1]

12 Jan 2024

Topological defects in self-assembled patterns of mesenchymal stromal cells in vitro are predictive attributes of condensation and chondrogenesis

PONE-D-23-21148R1

Dear Dr. Van Vliet,

We’re pleased to inform you that your manuscript has been judged scientifically suitable for publication and will be formally accepted for publication once it meets all outstanding technical requirements.

Kind regards,

Nileshkumar Dubey

Academic Editor

PLOS ONE
---

## [Editor Report · Acceptance letter]

18 Mar 2024

PONE-D-23-21148R1 

PLOS ONE

Dear Dr. Van Vliet, 

I'm pleased to inform you that your manuscript has been deemed suitable for publication in PLOS ONE. Congratulations! Your manuscript is now being handed over to our production team.

Kind regards, 

on behalf of

Dr. Nileshkumar Dubey 

Academic Editor

PLOS ONE